# Fast Elastic-Net Multi-view Clustering: A Geometric Interpretation Perspective

## ABSTRACT

Multi-view clustering methods have been extensively explored in the last decades. This kind of methods is built on the assumption that the data are sampled from multiple subspaces with low dimension and each group fits into one of these subspaces. The quadratic or cubic computation complexity produced by these methods is inevitable, resulting in the difficulty for clustering multi-view datasets with large scales. Some efforts have been presented to select key anchors beforehand to capture the data distributions in different views. Despite significant progress, these methods pay few attentions to deriving provably scalable and correct method for finding the optimal shared anchor graph from the geometric interpretation perspective. They also ignore to give a well balance between the connectedness and subspace preserving properties of the shared anchor graph. In this paper, we propose the Fast Elastic-Net Multi-view Clustering (FENMC) from a geometric interpretation perspective. We provide the geometric analysis in determining the optimal shared anchor graph based on the introduced elastic-net regularizer for fast multi-view clustering, where the elastic-net regularizer is built on the mixture of $L_2$ and $L_1$ norms. We also give a theoretical justification for the balance between the connectedness and subspace preserving properties of the shared anchor graph for multi-view clustering. Our experiments on different datasets show that the proposed method not only obtains the satisfied clustering performance, but also deals with large-scale datasets with high efficiency.

## CCS CONCEPTS

• **Computing methodologies → Artificial intelligence**.

## KEYWORDS

Geometric interpretation, elastic-net regularizer, multi-view clustering, connectedness, subspace preserving.

## 1 INTRODUCTION

In various computer vision tasks, including motion segmentation [5], image representation [11], feature extraction [7, 13, 18] and face clustering [10], the high-dimensional datasets can be approximated by a union subspaces with low dimensions. The subspace clustering [28, 33] has been widely explored in recent decades, which recovers the underlying structure of data with low dimensions and assigns

**Unpublished working draft. Not for distribution.**

each data point to the corresponding subspace. Significant progress has been made in uncovering such underlying subspace representations and subspace clustering based on graph usually produces the satisfied performance among multiple approaches. As a result, the subspace clustering based on graph has received great attention and a large number of methods have been presented.

Among these subspace clustering methods based on graph, some representative works are sparse subspace clustering [6], least squares regression (LSR) [20] and low-rank representation (LRR) [17]. They learn an $n \times n$ graph to represent the pairwise similarity between data points and then use the learned graph as the input to the existing clustering algorithm, i.e., spectral clustering. Moreover, some subspace clustering networks [12] have been developed to enjoy the merit of discriminative feature representations achieved by deep neural networks. It typically requires $O(n^2)$ to obtain the graph and $O(n^3)$ for eigen-decomposition in spectral clustering algorithm, where $n$ is the total number of data points in dataset. Thus, these two steps inevitably restrict the subspace clustering application on the datasets with large scales.

Recently, some research endeavors are devoted to accelerate subspace clustering [35]. For instance, Wang et al. [30] aimed to speed up the computation by adopting a data selection approach. You et al. [33] reduced the computation load based on the orthogonal matching pursuit. Peng et al. [22] converted the large-scale clustering issue as an out-of-sample problem. Alder et al. [1] combined the bipartite graph and sparse representation, resulting in a linear subspace clustering algorithm. Qin et al. [23] targeted at achieving an analytical, symmetrical and nonnegative similarity matrix for dealing with the data with large scales. Unfortunately, these accelerated subspace clustering methods usually target for the scenario with single view and fail to deal with the multi-view data.

The heterogeneous feature representations usually provide complementary information [37], i.e., a video might includes text, sounds and images [32]. As a result, multi-view subspace clustering methods have been exploited [2, 34]. For example, Cao et al. [2] studied both the diversity and consistency among different views. Zhang et al. [34] employed the latent space to perform subspace clustering. Qin et al. [24] explicitly extended the existing multi-view clustering in a semi-supervised manner and used small amount of supervisory information to construct an anti-block-diagonal indicator matrix. Compared with the methods for single view, these methods usually produce more desired results.

Unfortunately, most existing multi-view subspace clustering methods encounter the scalability issue, which limits their real application on the dataset with large scales. Some multi-view clustering approaches for dealing with large-scale data have been developed [26]. For example, Han et al. [9] reduced the number of matrix multiplication in the optimization procedure by regarding the intermediate factor matrix as a matrix with diagonal structure. Kang et al. [15] studied smaller graphs for multiple views based on

the built anchor bases. Wang et al. [29] explored the shared anchor graph in multi-view subspace clustering, which is guided by the consensus anchor bases in the data. Sun et al. [27] employed the underlying data distribution to learn anchor graph. Chen et al. [3] jointly obtained a more flexible and discriminative anchor representation and the cluster indicator with linear complexity. Despite significant progress, these methods pay few attentions to deriving a scalable and correct method for finding the optimal shared anchor graph in a provable manner from the geometric interpretation perspective. They also ignore to provide a well balance between the connectedness and subspace preserving properties of the shared anchor graph in multi-view subspace clustering.

In this work, we propose a Fast Elastic-Net Multi-view Clustering (FENMC) from a geometric interpretation perspective to address the above two issues. To be specific, we introduce the elastic -net regularizer built on the mixture of $L_2$ and $L_1$ norms for the learned shared anchor graph in multi-view subspace clustering, where $L_2$ norm improves the connectivity and $L_1$ helps obtain a subspace preserving affinity. The geometry of the elastic-net regularizer is explored and we then adopt it for deriving a provably scalable and correct method in achieving the optimal shared anchor graph. Our analysis shows a geometric interpretation and theoretical justification for the balance between the connectedness and subspace preserving properties of the shared anchor graph in multi-view subspace clustering.

The major contributions of this work include

- We propose a Fast Elastic-Net Multi-view Clustering (FENMC) from a geometric interpretation perspective. We provide the geometric analysis in determining the optimal shared anchor graph based on the introduced elastic-net regularizer for fast multi-view clustering, where the elastic-net regularizer is built on the mixture of $L_2$ and $L_1$ norms as well as a refined-anchor algorithm is designed to achieve further efficiency.
- We provide a geometric interpretation and theoretical justification for the balance between the connectedness and subspace preserving properties of the shared anchor graph based on the elastic-net regularizer for multi-view clustering.
- We conduct extensive experiments on several multi-view datasets to show that the proposed method is able to obtain the satisfied clustering performance and handle large-scale datasets with high efficiency in terms of different metrics.

## 2 RELATED WORK

As an efficient way, anchors or landmarks are adopted for scalable clustering on large-scale datasets. It usually selects relatively smaller number of data points termed anchors or landmarks to denote the neighborhood structure in the dataset. To be specific, we can build a small graph $S \in R^{m \times n}$ to measure the relationship between the entire dataset and the anchors with the guidance of $m$ anchors $A = \{a_1, \cdots, a_m\} \in R^{d \times m}$. The commonly used Gaussian kernel function can be employed to construct the graph $S$. However, it is not flexible enough in characterizing the complex data.

We can treat $A$ as a dictionary and learn affinity matrix $S$ for subspace clustering as follows:

$$\min_S \|X - AS\|_F^2 + \eta \|S\|_F^2, \quad s.t. \ S \geq 0, \ S\mathbf{1} = 1, \tag{1}$$

where $\eta > 0$ represents the balance parameter and $\mathbf{1}$ is a vector with all entries being one. It is observed that the above approach for constructing the graph is extremely efficient since the computation complexity is low. Likewise, the above model can be extended to the case for dealing with multi-view datasets and some recent works have incorporated anchor graphs into multi-view clustering. Liu et al. [19] combined graph construction and anchor learning for boosting clustering performance and imposed a graph connectivity constraint in the learning process. Yang et al. [31] used the multiple anchor graphs to achieve the efficient $K$-means clustering on multi-view dataset. However, the existing methods pay few attentions to deriving a scalable and correct method for finding the optimal shared anchor graph in a provable way from the geometric interpretation perspective. These methods also ignore to give a well balance between the connectedness and subspace preserving properties of the shared anchor graph in multi-view subspace clustering.

## 3 THE PROPOSED METHOD

In this part, we describe the proposed method in details, which includes the motivation, formulation and the complexity analysis.

### 3.1 Motivation and Formulation

There is much redundancy in the multi-view dataset with large scales and a small number of data points are enough to reconstruct the underlying subspaces. The smaller matrix $S \in R^{m \times n}$ can be adopted to approximate the full matrix, where $m \ll n$. That is, $S$ is achieved based on the anchors $A \in R^{d \times m}$ and the dataset $X^p$ for the $p$-th view. However, the existing multi-view clustering methods pay few attentions to deriving a scalable and correct method for finding the optimal shared anchor graph in a provable way from the geometric interpretation perspective. These methods also ignore to give a well balance between the connectedness and subspace preserving properties of the shared anchor graph in multi-view subspace clustering.

Based on the assumption that a consensus subspace with low dimension is shared by the high-dimensional data from different views, the learned anchors are expected to be consistent in the consensus space. Given multi-view dataset $\{X^p \in R^{d_p \times n}\}_{p=1}^v$ with $d_p$ and $n$ being the dimension and size of dataset, we define the projection matrix $U^p$ and then align the consensus anchors $A$ for the $p$-th view. To bridge the gap between the connectedness and subspace preserving properties of the obtained anchor graph $S$, we introduce a mixed $L_1$ and $L_2$ norm $r(.)$ to the anchor graph. Here, the mixed norm is also called the elastic-net regularizer. The above process is formulated as

$$r(s) = \lambda \|s\|_1 + \frac{1 - \lambda}{2} \|s\|_2^2, \tag{2}$$

where $\lambda \in [0, 1]$ is the trade-off parameter for the two regularizers. Thus, the proposed method has the formulation as follows:

$$\min_{U^p, A, S, \alpha} \sum_{p=1}^{v} \alpha_p^2 ||X^p - U^p AS||_F^2 + \lambda ||S||_1 + \frac{1-\lambda}{2} ||S||_2^2,$$ (3)

$$s.t. \ S \geq 0, \ S^T \mathbf{1} = 1, \ (U^p)^T U^p = I, \ A^T A = I,$$

where $\alpha_p^2 > 0$ represents the parameter of weight coefficient to learn. The above optimization problem can be solved by solving each variable while fixing the others. For simplicity, we adopt the objective function $f(s; U^p A)$ to denote the optimization problem in Eq. (3) and omit the constraints of variables in this formulation for the $p$-th view, defined as

$$f(s; U^p A) := \alpha_p^2 ||x^p - U^p As||_F^2 + \lambda ||s||_1 + \frac{1-\lambda}{2} ||s||_2^2.$$ (4)

Without loss of generality, $\{u_j^p a_j\}_{j=1}^n$ and $x^p$ are assumed to be normalized in the manner of unit $L_2$ norm. The above model then calculates

$$s^*(U^p A) := \arg \min_s f(s; U^p A).$$ (5)

For clarity, we adopt the notation $s^*$ in place of $s^*(U^p A)$ and fix $\alpha$ in Eq. (4) for the following analysis. Due to the convex property of $f(s; U^p A)$, we can guarantee that the obtained $s^*(U^p A)$ is unique. In the following part, we provide the detailed analysis of the solution to the proposed method from a geometric interpretation perspective and then design a refined-anchor algorithm to achieve further efficiency. We first give the concept of the trigger point in the following.

**Definition 1.** (*Trigger Point*) The trigger point regarding optimization problem Eq. (5) is

$$\vartheta(U^p A) := \varrho(x^p - U^p As^*(U^p A)),$$ (6)

where $\varrho > 0$ is the parameter. We adopt $\vartheta$ to represent $\vartheta(U^p A)$ for simplicity and find that the trigger point can be calculated when the optimal $s^*$ is obtained. Likewise, the solution $s^*$ can be directly achieved once the trigger point $\vartheta$ is known, which is shown in the theorem as follows.

**Theorem 1.** The optimal $s^*$ to Eq. (5) satisfies

$$(1 - \lambda)s^* = \nabla_\lambda((U^p A)^T \vartheta),$$ (7)

where $\nabla_\lambda(.)$ represents the soft-thresholding operator. It is defined as 0 if $|(U^p A)^T \vartheta| \leq \lambda$ and $sgn((U^p A)^T)(|(U^p A)^T| - \lambda)$ otherwise.

**Proof.** Due to the convex property of Eq. (5), $s^*$ is unique optimal if and only if it satisfies the optimally condition based on the partial derivative value regarding $s^*$ as:

$$(U^p A)^T \varrho(x^p - U^p As^*) = (1 - \lambda)s^* + \lambda z,$$ (8)

where $z \in \partial ||s^*||_1$. We then take the soft-thresholding on Eq. (8) for both sides by $\nabla_\lambda(.)$. Thus, Eq. (7) in Theorem 1 can be obtained. The reverse implication can be proved by establishing that the $j$-th row of Eq. (8) is satisfied when the corresponding row in Eq. (7) holds for three cases $s^* < 0$, $s^* = 0$ and $s^* > 0$ separately.

As shown in Theorem 1, the value of $s^*$ is determined by the angle between $u_j^p a_j$ and $\vartheta$. The inequation $|\langle u_j^p a_j, \vartheta \rangle| \leq \lambda$ holds when $u_j^p a_j$ is far from $\vartheta$ to certain degree, resulting in that $s_j^*$ is equal to zero. We call the region $s^* \neq 0$ as the trigger region and

use the quality $\varphi(v^p, \vartheta) := \frac{|\langle v^p, \vartheta \rangle|}{||v||_2 ||\vartheta||_2}$ for representing the coherence. The trigger region is formally defined as follows.

**Definition 2.** (*Trigger Region*) The trigger region for the optimization problem Eq. (5) is defined as

$$\Gamma(U^p A) := \{v^p \in R^{d_p} : ||v^p||_2 = 1, \ \varphi(v^p, \vartheta) > \frac{\lambda}{||\vartheta||_2}\}.$$ (9)

According to Theorem 1 and the above definition, we can obtain that $s^* \neq 0$ if and only if $u_j^p a_j \in \Gamma(U^p A)$. The properties of solution can be captured by the trigger region when new columns are added or columns are removed from $U^p A$, which gives us the key insight in designing the refined-anchor method. The basic measure is for solving the reduced-scale problem determined from the trigger region and the obtained anchor is called refined anchor. We denote the refined anchor at iteration $i$ as $T_i$ and select the next refined anchor $T_{i+1}$ from the trigger region $\Gamma(U_{T_i}^p A_{T_i})$, where $U_{T_i}^p$ and $A_{T_i}$ denote the submatrix of $U^p$ and $A$ with columns indexed by $T_i$, respectively. This iterative process is terminated when $T_{i+1}$ no longer contains any new data points. To show the convergence of this refined-anchor method, we give the lemma as follows.

**Lemma 1.** If $T_{i+1} \nsubseteq T_k$, then

$$f(s^*(U_{T_{i+1}}^p A_{T_{i+1}}); U_{T_{i+1}}^p A_{T_{i+1}}) < f(s^*(U_{T_i}^p A_{T_i}); U_{T_i}^p A_{T_i}).$$ (10)

**Proof.** We first define the sets

$$L := T_{i+1} \setminus T_i \neq \emptyset, \ Q := T_i \setminus T_{i+1}, \ J := T_i \cap T_{i+1}.$$ (11)

Based on these definitions, we can obtain $T_i = Q \cup J$ and $T_{i+1} = J \cup L$. Since $T_{i+1}$ consists of columns of $U^p A$ in $\Gamma(U_{T_i}^p A_{T_i})$, we can achieve that there is no column of $U_Q^p A_Q$ in $\Gamma(U_{T_i}^p A_{T_i})$. Considering $U_{T_i}^p A_{T_i} = [U_J^p A_J, U_Q^p A_Q]$, we have

$$\begin{aligned} f(s^*(U_{T_i}^p A_{T_i}); U_{T_i}^p A_{T_i}) &= f([\mathbf{0}, s^*(U_J^p A_J)]^T; [U_J^p A_J, U_L^p A_L]) \\ &\geq \min_s f(s; [U_J^p A_J, U_L^p A_L]) \\ &= f(s^*([U_J^p A_J, U_L^p A_L]); [U_J^p A_J, U_L^p A_L]) \\ &= f(s^*(U_{T_{i+1}}^p A_{T_{i+1}}); U_{T_{i+1}}^p A_{T_{i+1}}). \end{aligned}$$ (12)

We then achieve Theorem 2 with the guidance of Lemma 1, which is shown in the following.

**Theorem 2.** The refined-anchor algorithm converges to the optimal $s^*(U^p A)$ in a finite number of iterations.

**Proof.** We find that the objective function is guaranteed to be decreasing for each iteration before the termination happens according to Lemma 1. Since there are limited number of entries in $T$, we conclude that the refined-anchor algorithm converges in a finite number of iterations with $T_{i+1} \subset T_i$. We construct $s^*$ such that $s_{T_i^s}^* = 0$ when $T_i^s$ is the complement of $T_i$ and $s_{T_i}^* = s^*(U_{T_i}^p A_{T_i})$ otherwise. Due to $(1 - \lambda)s^* = \nabla_\lambda((U_{T_i}^p A_{T_i})^T \vartheta)$ in Theorem 1, we have $\nabla_\lambda((u_j^p a_j)^T \vartheta(U_{T_i}^p A_{T_i})) = 0$. Thus, the $s^*(U^p A)$ is optimal when the refined-anchor algorithm converges.

The refined-anchor algorithm can effectively deal with the large-scale problems, which solves the reduced-size problems by updating $s^*(U_{T_i}^p A_{T_i})$ in Algorithm 1. We then provide conditions for the

**Algorithm 1:** The refined-anchor algorithm

**Input:** $U^p$, $A$, $\alpha$ and $\lambda$.

**Output:** The optimal $s^*$.

**Initialize:** Initialize $T_0$ and set $i = 0$.

**repeat**

    Update $s^*(U^p_{T_i} A_{T_i})$ in Eq. (5);

    Update $\vartheta(U^p_{T_i} A_{T_i})$ in Eq. (6);

    Update $T_{i+1} = \{j : u^p_j a_j \in \Gamma(U^p_{T_i} A_{T_i})\}$;

    $i = i + 1$.

**until** $T_{i+1} \subseteq T_i$;

shared anchor graph to be subspace preserving by balancing connectedness and subspace preserving properties. As Eq. (5), we calculate $s^*(u^p_j a_j, (U^p A)_{-j})$ for each $\{u^p_j a_j\}_{j=1}^n$, where $s^*(u^p_j a_j, (U^p A)_{-j})$ equals to $\arg\min_s f(s; u^p_j a_j, (U^p A)_{-j})$ and $(U^p A)_{-j}$ is $U^p A$ with $j$-th column removed. We can obtain that $s^*(u^p_j a_j, (U^p A)_{-j})$ is subspace preserving if there are no connections built between $u^p_j a_j$ and $(U^p A)_{-j}$ from different subspaces. The nonzero entries in $s^*(u^p_j a_j, (U^p A)_{-j})$ are desired to be dense for guaranting that the affinity graph is well-connected. That is, the connectedness and subspace preserving properties are two conflicting goals. We then provide the detailed analysis of the tradeoff between the connectedness and subspace preserving properties from a geometric interpretation perspective and the sufficient conditions when the anchor graph is subspace preserving.

We perform the detailed analysis upon the optimization problem $\min_s f(s; u^p_j a_j, (U^p A)^l_{-j})$, where $(U^p A)^l_{-j}$ denotes $(U^p A)^l$ in the $\Xi_l$ subspace with the $j$-th column removed. We regard anchors from other subspaces as newly added columns to $(U^p A)^l_{-j}$ and achieve the geometric result as follows.

**Theorem 3.** Supposing $u^p_j a_j \in \Xi_l$, then $s^*(u^p_j a_j, (U^p A)_{-j})$ is subspace preserving if and only if $x_e \notin \Gamma(u^p_j a_j, (U^p A)^l_{-j})$ for all $x_e \notin \Xi_l$.

**Proof.** According to the notation as above, $\Gamma(u^p_j a_j, (U^p A)^l_{-j})$ is the trigger region. We know that adding more data points that are not in the trigger region $\Gamma(u^p_j a_j, (U^p A)^l_{-j})$ does not affect the corresponding solution. Specifically, we have $s^*(u^p_j a_j, (U^p A)_{-j}) = P \cdot [s^*(u^p_j a_j, (U^p A)^l_{-j})^T, O^T]^T$ if $x_e \notin \Gamma(u^p_j a_j, (U^p A)^l_{-j})$ for all $x_e \notin \Xi_l$, where $P$ denotes some permutation matrix. Moreover, the vector of $s^*(u^p_j a_j, (U^p A)_{-j})$ corresponding to the data points outside of $\Xi_l$ is nonzero if any $x_e \notin \Xi_l$ in the trigger region $\Gamma(u^p_j a_j, (U^p A)^l_{-j})$. Thus, the obtained solution is incorrect in determining the $l$-th space.

The above theorem shows that $s^*(u^p_j a_j, (U^p A)_{-j})$ is subspace preserving if and only if all data points from other subspaces lie outside the trigger region. We desire a small trigger region to ensure that the solution is subspace preserving, while need a large trigger region to ensure the connectedness. It further highlights the tradeoff between connectedness and subspace preserving properties. The elastic-net promotes sparse and dense solution by $L_1$ and $L_2$ regularizations. Thus, as $\lambda$ increases from 0 towards 1, one should expect that the trigger region decreases in size.

To solve the optimization problem in Eq. (3), we adopt an alternate minimizing algorithm to optimize each variable with others being fixed.

## 3.2 Optimization

$S$-**subproblem**: By fixing the other variables, the objective function regarding $S$ is

$$\min_S \sum_{p=1}^v \alpha_p^2 ||X^p - U^p A S||_F^2 + \lambda ||S||_1 + \frac{1-\lambda}{2} ||S||_2^2, \tag{13}$$

$$s.t.\ S \geq 0,\ S^T \mathbf{1} = 1.$$

We then rewrite it as the Quadratic Programming (QP) problem as follows:

$$\min \frac{1}{2} S^T_{:,j} W S_{:,j} + h^T S_{:,j},\ s.t.,\ S \geq 0,\ S^T_{:,j} \mathbf{1} = 1, \tag{14}$$

where $h^T = -2 \sum_{p=1}^v (X^p_{:,j})^T U^p A + \lambda \partial ||s^*||_1$ and $W = 2(\sum_{p=1}^v \alpha_p^2 + \frac{1-\lambda}{2})I$.

$U^p$-**subproblem**: By fixing the other variables, the objective function regarding $U^p$ is

$$\min_{U^p} \sum_{p=1}^v \alpha_p^2 ||X^p - U^p A S||_F^2,\ s.t.\ (U^p)^T U^p = I. \tag{15}$$

We then transform it into the following form:

$$\max_{U^p} Tr((U^p)^T G^p),\ s.t.\ (U^p)^T U^p = I. \tag{16}$$

where $G^p = X^p S^T A^T$. The optimal $U^p$ is $WV^T$, where $W$ and $V$ are singular matrices for $G^p$.

$A$-**subproblem**: By fixing the other variables, the objective function regarding $A$ is

$$\min_A \sum_{p=1}^v \alpha_p^2 ||X^p - U^p A S||_F^2,\ s.t.\ A^T A = I. \tag{17}$$

Likewise, we can obtain the formulation as follows:

$$\max_A Tr(A^T E),\ s.t.\ A^T A = I, \tag{18}$$

where $E = \sum_{p=1}^v \alpha_p^2 (U^p)^T X^p S^T$. The optimal $A$ can be obtained by $\Delta \Gamma^T$, where $\Delta$ and $\Gamma^T$ are singular matrices for $E$.

$\alpha_p$-**subproblem**: By fixing the other variables, the objective function regarding $F$ is

$$\min_\alpha \sum_{p=1}^v \alpha_p^2 M_p,\ s.t.\ \alpha^T \mathbf{1} = 1, \tag{19}$$

where $M_p = ||X^p - U^p A S||_F^2$. According to the Cauchy-Buniakowsky-Schwarz inequality, the optimal $\alpha_p$ is

$$\alpha_p = \frac{\frac{1}{M_p}}{\sum_{p=1}^v \frac{1}{M_p}}. \tag{20}$$

Due to the optimal solution and convex property of each subproblem, the objective function monotonically decreases in each iteration until convergence. We summarize the whole process for solving the proposed FENMC in Algorithm 2. Then the refined-anchor algorithm as shown in Algorithm 1 can be adopted to achieve further efficiency.

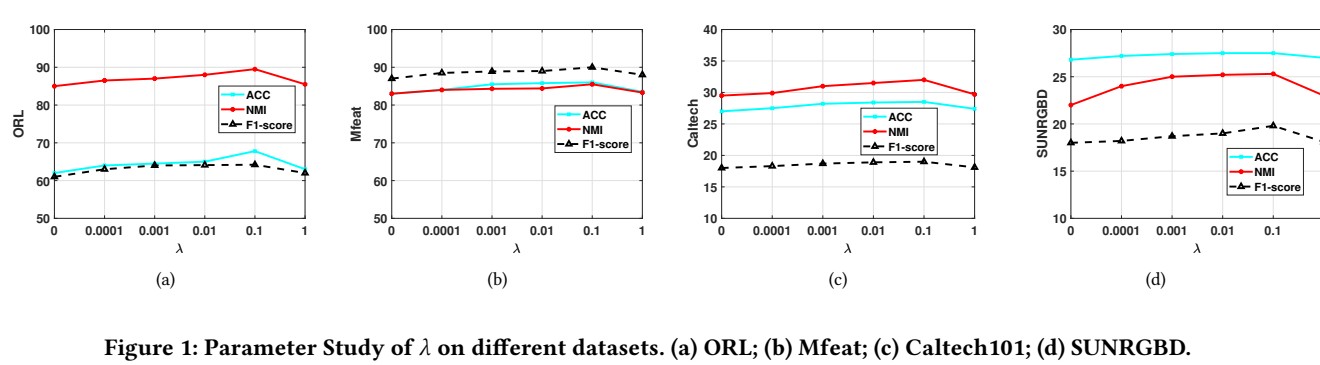

Figure 1: Parameter Study of $\lambda$ on different datasets. (a) ORL; (b) Mfeat; (c) Caltech101; (d) SUNRGBD.

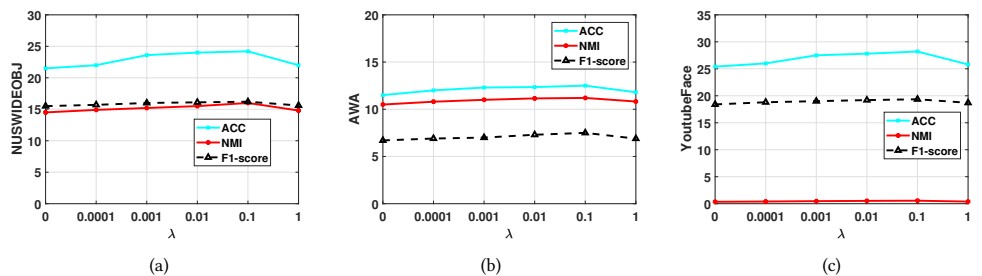

Figure 2: Parameter Study of $\lambda$ on different datasets. (a) NUSWIDEOBJ; (b) AWA; (c) YoutubeFace.

Table 1: Clustering results based on ACC (%) on all datasets. "N/A " denotes out of memory.

| Dataset | AMGL | SFMC | BMVC | LMVSC | MSGL | FPMVS | EOMSC-CA | OMSC | Ours |
|---|---|---|---|---|---|---|---|---|---|
| ORL | 69.50±0.01 | 61.40±0.05 | 48.70±0.05 | 58.60±0.02 | 21.00±0.05 | 52.00±0.50 | 62.20±0.05 | 63.80±0.00 | 68.00±0.30 |
| Mfeat | 82.60±0.02 | 75.50±0.20 | 69.30±0.05 | 81.75±0.05 | 75.40±0.02 | 82.20±0.05 | 82.25±0.03 | 84.00±0.05 | 86.00±0.10 |
| Caltech101 | 14.80±0.01 | 17.70±0.05 | 21.20±0.03 | 15.50±0.01 | 14.10±0.02 | 28.50±0.05 | 22.30±0.03 | 24.00±0.00 | 28.50±0.00 |
| SUNRGBD | 9.80±0.01 | 11.30±0.05 | 16.70±0.01 | 18.00±0.05 | 13.00±0.01 | 23.40±0.05 | 23.70±0.05 | 25.20±0.00 | 27.50±0.20 |
| NUSWIDEOBJ | N/A | 12.20±0.05 | 12.90±0.05 | 14.70±0.05 | 12.00±0.05 | 19.20±0.05 | 19.60±0.05 | 21.00±0.05 | 24.20±0.50 |
| AWA | N/A | 3.92±0.03 | 8.60±0.05 | 7.20±0.03 | 8.00±0.02 | 8.90±0.01 | 8.65±0.05 | 9.00±0.10 | 12.50±0.00 |
| YoutubeFace | N/A | N/A | 8.90±0.05 | 14.00±0.02 | 16.70±0.01 | 23.00±0.03 | 26.45±0.05 | 26.50±0.00 | 28.20±0.00 |

Table 2: Clustering results based on NMI (%) on all datasets. "N/A " denotes out of memory.

| Dataset | AMGL | SFMC | BMVC | LMVSC | MSGL | FPMVS | EOMSC-CA | OMSC | Ours |
|---|---|---|---|---|---|---|---|---|---|
| ORL | 87.10±0.07 | 82.70±0.01 | 67.70±0.03 | 78.50±0.03 | 43.70±0.02 | 74.40±0.05 | 88.10±0.02 | 88.50±0.10 | 90.00±0.60 |
| Mfeat | 86.70±0.05 | 86.80±0.10 | 66.05±0.15 | 76.00±0.20 | 76.54±0.05 | 79.40±0.01 | 83.20±0.15 | 84.20±0.10 | 85.50±0.10 |
| Caltech101 | 35.30±0.01 | 26.10±0.03 | 42.50±0.04 | 33.30±0.02 | 26.10±0.02 | 34.10±0.05 | 24.65±0.05 | 30.00±0.00 | 32.00±0.10 |
| SUNRGBD | 18.50±0.10 | 2.30±0.05 | 19.50±0.05 | 25.50±0.05 | 9.30±0.05 | 24.10±0.05 | 22.50±0.01 | 24.30±0.00 | 25.30±0.25 |
| NUSWIDEOBJ | N/A | 0.96±0.01 | 12.90±0.02 | 12.80±0.05 | 5.70±0.03 | 13.20±0.05 | 13.20±0.15 | 14.00±0.00 | 16.00±0.50 |
| AWA | N/A | 0.30±0.05 | 13.70±0.02 | 8.50±0.05 | 7.90±0.03 | 10.50±0.03 | 9.70±0.03 | 10.00±0.02 | 11.20±0.10 |
| YoutubeFace | N/A | N/A | 5.90±0.05 | 11.80±0.01 | 0.07±0.01 | 2.40±0.01 | 0.32±0.01 | 0.37±0.00 | 0.55±0.05 |

## 3.3 Complexity Analysis

FENMC has a relatively low computation complexity since the adopted anchor strategy. To be specific, it needs $O(nm^2 + m^3 + nmd + nm\sum_{p=1}^{v} d_p)$ in updating $S$. In optimizing $U^p$, the SVD consumes $O(d_p d^2)$ and matrix multiplication takes $O(md_p(n+d))$ for

each view. It needs $O(md^2)$ in SVD and $O(nd(m + d_p))$ in matrix multiplication for updating $A$. In updating $\alpha_p$, it takes $O(1)$. With the obtained shared anchor graph $S$, we conduct a linear graph algorithm and then adopt $K$-means to obtain the results and the corresponding computation cost is $O(nmd)$. The total time cost of

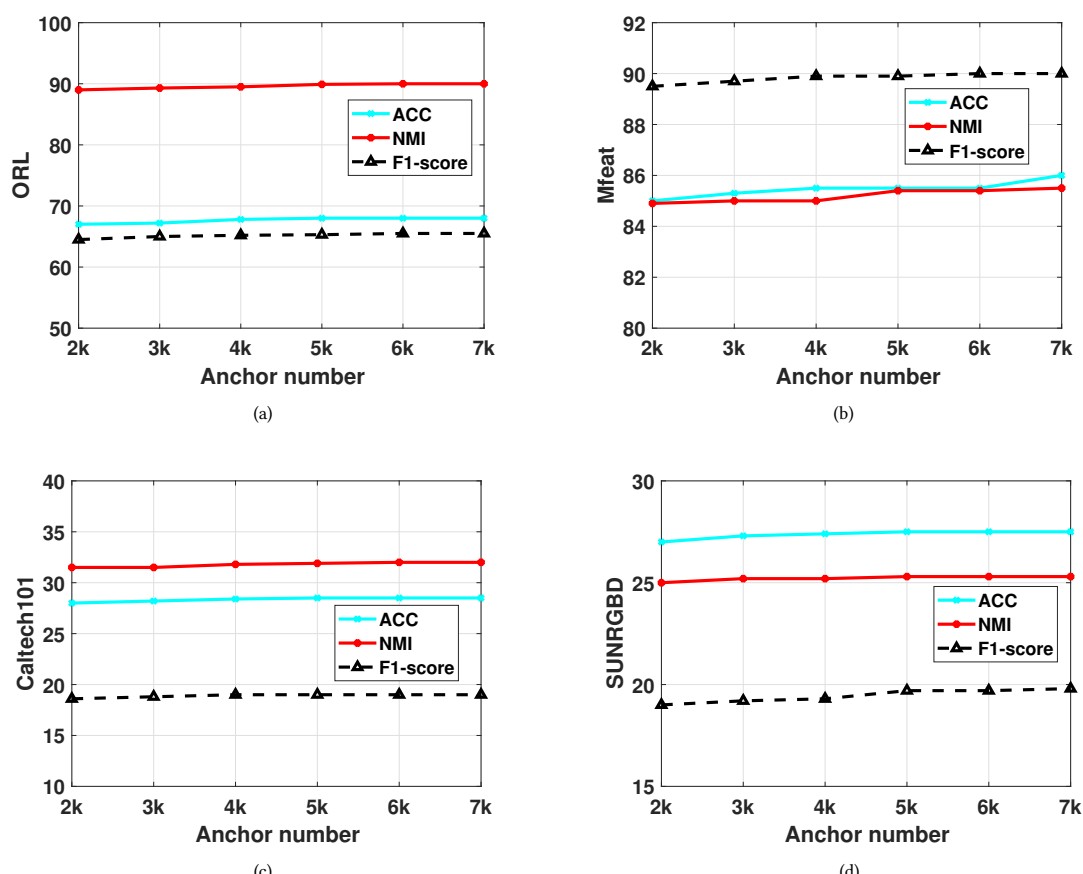

Figure 3: Sensity Study of anchor numbers on different datasets. (a) ORL; (b) Mfeat; (c) Caltech101; (d) SUNRGBD.

Table 3: Clustering results based on F1-score (%) on all datasets. "N/A " denotes out of memory.

| Dataset | AMGL | SFMC | BMVC | LMVSC | MSGL | FPMVS | EOMSC-CA | OMSC | Ours |
|---|---|---|---|---|---|---|---|---|---|
| ORL | 51.20±0.03 | 30.60±0.05 | 30.50±0.04 | 45.90±0.09 | 51.50±0.20 | 38.40±0.15 | 62.10±0.00 | 63.20±0.10 | 65.50±0.40 |
| Mfeat | 80.90±0.05 | 71.10±0.15 | 58.80±0.01 | 72.50±0.02 | 70.10±0.02 | 76.00±0.40 | 77.00±0.01 | 78.20±0.00 | 90.00±0.20 |
| Caltech101 | 4.05±0.10 | 4.65±0.10 | 18.50±0.05 | 10.50±0.05 | 8.60±0.04 | 20.90±0.03 | 10.80±0.03 | 15.00±0.00 | 19.00±0.50 |
| SUNRGBD | 6.40±0.40 | 12.10±0.00 | 10.20±0.01 | 11.60±0.20 | 9.50±0.15 | 16.00±0.05 | 15.30±0.05 | 17.00±0.00 | 19.80±0.50 |
| NUSWIDEOBJ | N/A | 11.50±0.01 | 8.80±0.02 | 9.30±0.05 | 8.50±0.05 | 13.50±0.07 | 13.60±0.05 | 14.50±0.00 | 16.20±0.20 |
| AWA | N/A | 4.60±0.03 | 5.59±0.02 | 3.60±0.05 | 4.20±0.01 | 6.20±0.05 | 5.90±0.05 | 6.20±0.00 | 7.50±0.50 |
| YoutubeFace | N/A | N/A | 5.80±0.02 | 8.30±0.01 | 15.00±0.10 | 14.00±0.05 | 16.40±0.01 | 17.10±0.00 | 19.35±0.30 |

FENMC is $O((nm^2 + m^3 + nmd + nm \sum_{p=1}^{v} d_p + md_p(n+d) + d_p d^2 + nd(m+d_p) + md^2 + nmd)t)$, where $t$ denotes the iteration number. Since $n \gg m$ and $n \gg k$, the computation complexity of FENMC is almost linear to the number of data points.

## 4 EXPERIMENTS

We perform experiments in this part to demonstrate the performance of the proposed method in terms of effectiveness and efficiency on several datasets. We conduct all experiments on a standard Window PC with AMD Ryzen 5 1600X 3.60 GHz.

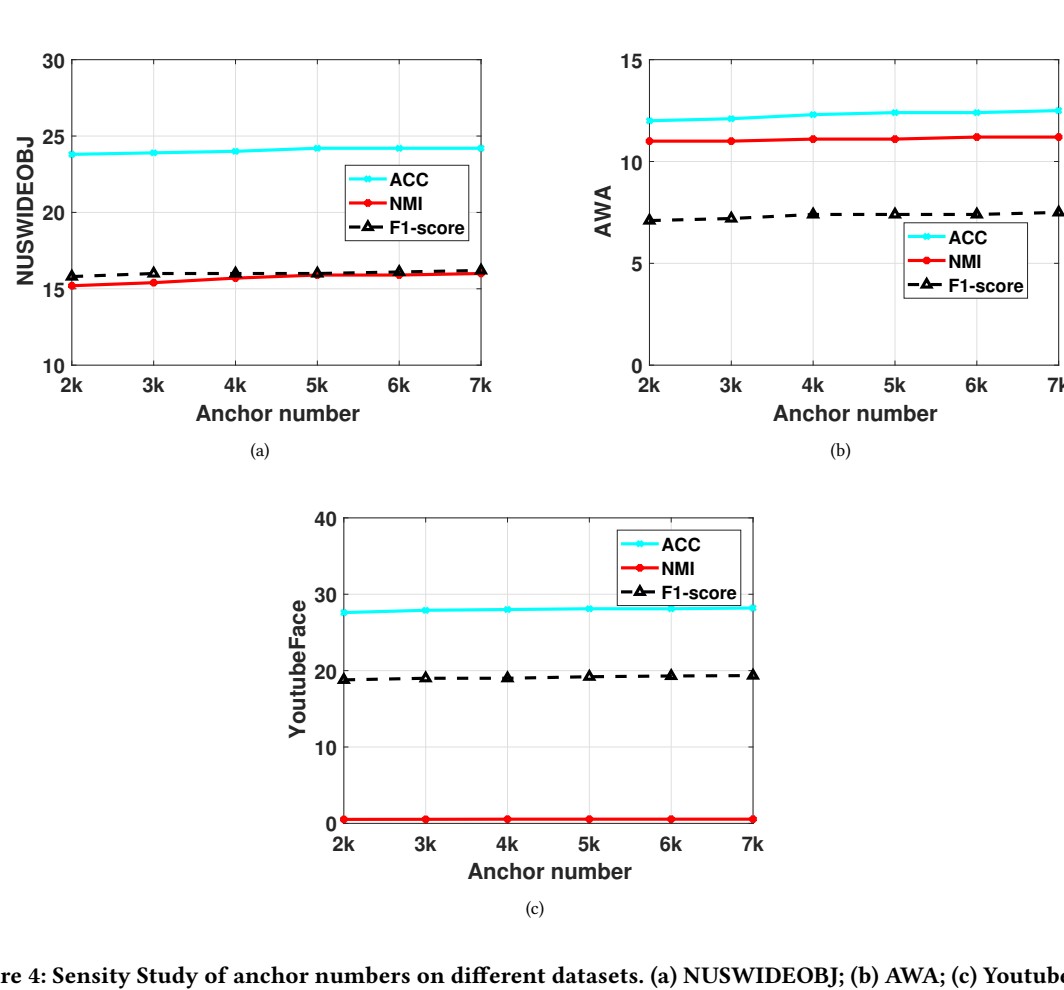

Figure 4: Sensity Study of anchor numbers on different datasets. (a) NUSWIDEOBJ; (b) AWA; (c) YoutubeFace.

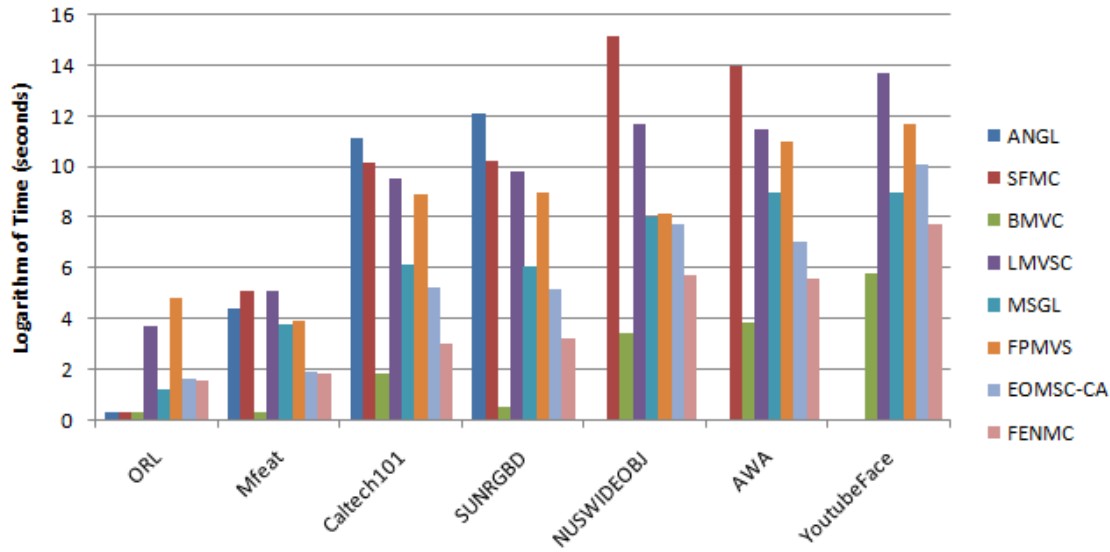

Figure 5: Logarithm of running time of on different datasets.

---

**Algorithm 2:** Algorithm of FENMC

**Input:** Multi-view dataset $\{X_p\}_{p=1}^{v}$, number of clusters $k$,
parameter $\lambda$.

**Output:** The shared anchor graph $S$.

**Initialize:** Initialize $U^p$, $A$, $\{\alpha_p\}_{p=1}^{v}$ and $S$.

**repeat**

| Update $S$ according to Eq. (14);
| Update $\{U_p\}_{p=1}^{v}$ according to Eq. (16);
| Update $A$ according to Eq. (18);
| Update $\alpha$ according to Eq. (20);

**until** *convergence*;

---

## 4.1 Benchmark Datasets

We conduct experiments on seven commonly adopted multi-view datasets, which includes AWA, Caltech101 [8], ORL, Mfeat, SUN-RGBD [25], NUSWIDEOBJ [4] and YoutubeFace. To be specific, AWA has total 30475 subjects originated from 50 classes. Caltech consists of 102 classes and 9144 subjects. ORL has 400 images and 40 classes. Mfeat is generated from UCI machine learning repository, which consists of the digits from 0 to 9. SUNRGBD has total 45 classes and 10335 indoor images. NUSWIDEOBJ is mainly used for object recognition, which contains 30000 objects. YoutubeFace is produced from YouTube and has total 101499 instances.

## 4.2 Experimental Settings

We compare the proposed method with eight multi-view clustering approaches including AMGL [21], BMVC [36], LMVSC [15], SFMC [16], MSGL [14], EOMSC-CA [19], OSMC [3] and FRMVS [29].

We need to determine the anchor number in the experiment. Since the number of data points adopted for reflecting the underlying subspaces is expected to be not less than the total number of subspaces, we tune the anchor number in the range of $[2k, 3k, \cdots, 7k]$, where $k$ denotes the cluster number in dataset. To guarantee the fair comparison, we adopt the experimental settings stated in the corresponding compared methods and use the best parameters for them. We select $\lambda$ from the range $[0, 0.0001, 0.001, 0.01, 0.1, 1]$, which influences the connectedness and subspace preserving properties of the proposed method. To evaluate the performance of all methods, we employ accuracy (ACC), normalized mutual information (NMI) and F1-score in the experiment.

## 4.3 Parameter Selection

In this section, we analyze how the parameter $\lambda$ influences the clustering performance of the proposed method on different datasets in terms of ACC, NMI and F1-score. It is selected in the range $[0, 0.0001, 0.001, 0.01, 0.1, 1]$ and the impacts leaded by the parameter is given. Note that $\lambda = 1$ and $\lambda = 0$ also correspond to the ablation studies when connectedness and subspace preserving properties are not considered, respectively. Based on Figs. 1-2, we observe that desired clustering performance is achieved when $\lambda = 0.1$, which demonstrates that simultaneously considering connectedness and subspace preserving properties for the proposed method with a appropriate tradeoff is helpful for achieving a desired shared anchor graph. Moreover, connectedness and subspace preserving

properties are both important and should be simultaneously considered for the obtained anchor graph.

## 4.4 Sensity Study

We study how the anchor number impacts the clustering performance under ACC, NMI and F1-score in this part. The sensity anaysis is performed on different datasets regarding the anchor number under these metrics. Based on Figs. 3-4, we find that generally stable performance can be produced with the varying number of anchors on these datasets, which validates that the anchor number does not play a vital important role in guiding desired performance for the proposed method.

## 4.5 Experimental Results

We give the clustering results of the proposed method and other methods for comparison on different datasets in Tables 1-3. In the experiment, N/A is adopted to represent the out-of-memory issue for clarity. We repeat each experiment for 20 times and give the mean values as well as the standard deviations. Based on the achieved results, we can draw conclusions in the following:

- The proposed method achieves more desired clustering performance on most of the multi-view datasets, especially on the datasets with relatively large scales. For example, the proposed method is able to generate 14.4% improvements than MSGL on Caltech101 for the clustering results in terms of ACC.
- Methods based on the anchor tend to produce better results than the traditional multi-view clustering approaches for most cases, indicating that using anchors is critical to achieve satisfied graph on different datasets.
- The proposed method is able to behave better than other compared methods built on anchor, demonstrating that simultaneously taking connectedness and subspace preserving properties for the proposed method into consideration controlled by a proper tradeoff is beneficial to achieve an expected shared anchor graph.

## 4.6 Running Time

In this part, we show the running times consumed by all methods on different multi-view datasets. According to Fig. 5, we observe that relatively less running time is needed by the proposed method compared with some multi-view clustering approaches on most datasets, which can be explained by the fact that the necessity of constructing the shared anchor graph with relatively smaller size guided by the proposed method.

## 5 CONCLUSION

In this paper, we introduce a Fast Elastic-Net Multi-view Clustering (FENMC) from a geometric interpretation perspective. The geometric analysis for determining the optimal shared anchor graph based on the introduced elastic-net regularizer is given for fast multi-view clustering. We then provide a theoretical justification for the balance between the connectedness and subspace preserving properties of the shared anchor graph in multi-view clustering. Experiments on several multi-view datasets demonstrate that the proposed method owns desired effectiveness and efficiency.

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
