# OpenReview forum: "Fast Elastic-Net Multi-view Clustering: A Geometric Interpretation Perspective"
_acmmm.org/ACMMM/2024/Conference — MM2024 Poster_

### Official Review · Reviewer_HkMQ · 2024-05-15

**Rating:** 6
**Confidence:** 4

**Summary:**

A Fast ElasticNet Multi-view Clustering (FENMC) from a geometric interpretation perspective is designed in this paper. The authors provide the geometric analysis for fast multi-view clustering, where the elastic-net regularizer is built on the mixture of 𝐿2 and 𝐿1 norms. They also give a theoretical justification for the balance between the connectedness and subspace preserving properties of the shared anchor graph for multi-view clustering. Experiments on different datasets show that the proposed method not only obtains the satisfied clustering performance, but also deals with large-scale datasets with high efficiency.

**Strengths:**

1. The authors give the geometric analysis in determining the optimal shared anchor graph based on the introduced elasticnet regularizer for fast multi-view clustering.

2. Experiments on different datasets show that the proposed method not only obtains the satisfied clustering performance, but also deals with large-scale datasets with high efficiency.

**Limitations:**

1. The authors need to give the device memory in the experiment for the proposed FENMC.

2. The reason for the anchor number selection can be further analyzed in the experiment.

3. Why the parameter \lambda is selected as 0.0001, 0.001, 0.01, 0.1 and 1 for parameter study in the experiment.

**Suitability:**

3

---

### Official Review · Reviewer_Asuk · 2024-05-19

**Rating:** 5
**Confidence:** 3

**Summary:**

This paper proposes a Fast Elastic-Net Multi-view Clustering (FENMC) from a geometric interpretation perspective. It provides the geometric analysis in determining the optimal shared anchor graph based on the introduced elastic-net regularizer for fast multi-view clustering, where the elastic-net regularizer is built on the mixture of 𝐿2 and 𝐿1 norms as well as a refined-anchor algorithm is designed to achieve further efficiency.

**Strengths:**

1. A geometric interpretation and theoretical justification for the balance between the connectedness and subspace preserving properties of the shared anchor graph based on the elastic-net regularizer for multi-view clustering are given in this paper.
2. The paper is well-organized and easy to be understood by the readers.
3. Extensive experiments on several multi-view datasets are conducted to show that the proposed method is able to obtain the satisfied clustering performance and handle large-scale datasets with high efficiency in terms of different metrics

**Limitations:**

1. The best results in Tables 1-3 can be bolded to better show the performance gains.
2. N/A in line 3 of Section 4.5 can be denoted by “N/A” for better clarity.
3. The second-best results in Tables 1-3 can also be highlighted to make the results more visible.

**Suitability:**

2

---

### Official Review · Reviewer_f5Xe · 2024-05-24

**Rating:** 5
**Confidence:** 3

**Summary:**

The paper focuses on the problem of multi-view clustering from a geometric interpretation standpoint, introducing a Fast Elastic-Net Multi-view Clustering (FENMC) method. The primary goal is to tackle the scalability challenges encountered by existing methods when clustering large-scale multi-view datasets. The FENMC method incorporates an elastic-net regularizer that combines L2 and L1 norms to achieve a balance between connectedness and subspace-preserving properties within the shared anchor graph. It offers a theoretical justification for this balance and presents experimental results on various datasets to demonstrate the performance and efficiency of the proposed method.

**Strengths:**

The paper presents several strengths in its approach to clustering large-scale multi-view datasets. Firstly, it integrates an elastic-net regularizer that balances L2 and L1 norms, leading to a geometrically grounded and scalable solution for constructing an optimal shared anchor graph. Secondly, the FENMC method provides a geometric interpretation and theoretical justification for the balance between connectedness and subspace preserving properties within the graph. Lastly, the method's clear formulation, complexity analysis, and experimental results contribute to its ease of implementation and improvements in clustering performance across various datasets of considerable scale.

**Limitations:**

The manuscript would benefit from a more detailed explanation of the geometric interpretation perspective and how the proposed method differs from existing approaches in finding the optimal shared anchor graph. Additionally, incorporating more recent anchor graph-based methods in the comparative experiments would provide a more comprehensive evaluation. Furthermore, the relationship between the theorems presented and the overall model needs to be clarified, specifying what each theorem proves and how these results support the proposed method. A thorough review and correction of the notation used throughout the manuscript are necessary to ensure consistency and readability.

**Suitability:**

2

---

### Official Review · Reviewer_gMVy · 2024-05-26

**Rating:** 6
**Confidence:** 3

**Summary:**

This paper proposes the Fast ElasticNet Multi-view Clustering (FENMC) from a geometric interpretation perspective. The authors provide the geometric analysis in determining the optimal shared anchor graph based on the introduced elasticnet regularizer for fast multi-view clustering, where the elastic-net regularizer is built on the mixture of 𝐿2 and 𝐿1 norms. The authors also give a theoretical justification for the balance between the connectedness and subspace preserving properties of the shared anchor graph for multi-view clustering.

**Strengths:**

1 The authors provide the geometric analysis in determining the optimal shared anchor graph based on the introduced elasticnet regularizer for fast multi-view clustering.

2. The authors also give a theoretical justification for the balance between the connectedness and subspace preserving properties of the shared anchor graph for multi-view clustering.

**Limitations:**

1. In the experimental part, the authors give the running time of methods on data sets, so the authors need to add the description of device memory in the experiment.

2. The subsection on the Running time contains relatively little information on the results of the experiments, and a little more description and analysis of the results could be added as appropriate.

3. The best performance in the table of experimental results can be bolded to make the results more visible.

4. The authors should confirm that the formats of references are consistent and correct.

**Suitability:**

2

---

### Meta-Review · Area_Chair_Surh · 2024-07-01

**Recommendation:** Accept (Poster)
**Confidence:** 5

**Metareview:**

This paper studies the scalability issue in anchor based multi-view clustering from a geometric interpretation perspective. It devises an elastic-net regularizer by combining L2 norm and L1 norm to receive an adaptive balance between connectedness and subspace-preserving properties. Then, it provides a theoretically reasonable and technically sound justification. After rebuttal and discussion, all reviewers are positive about this paper. So, acceptance is recommendated.